# Effects of Octenyl-Succinylated Chitosan—Whey Protein Isolated on Emulsion Properties, Astaxanthin Solubility, Stability, and Bioaccessibility

**DOI:** 10.3390/foods12152898

**Published:** 2023-07-30

**Authors:** Lingyu Han, Ruiyi Zhai, Bing Hu, Jixin Yang, Yaoyao Li, Zhe Xu, Yueyue Meng, Tingting Li

**Affiliations:** 1Key Laboratory of Biotechnology and Bioresources Utilization of Ministry of Education, College of Life Science, Dalian Minzu University, Dalian 116600, China; hly@dlnu.edu.cn (L.H.); 20201427@dlnu.edu.cn (B.H.); 18777440369@163.com (Y.L.); dlpuxz@163.com (Z.X.); 18945047510@163.com (Y.M.); 2Faculty of Arts, Science and Technology, Wrexham Glyndwr University, Plas Coch, Mold Road, Wrexham LL11 2AW, UK; jixinyang@hotmail.com

**Keywords:** chitosan oligosaccharide, octenyl succinic anhydride, astaxanthin, emulsions properties

## Abstract

The synthesis of octenyl-succinylated chitosan with different degrees of substitution resulting from chemical modification of chitosan and controlled addition of octenyl succinic acid was investigated. The modified products were characterized using ^1^H NMR, FTIR, and XRD, and the degree of substitution was also determined. The properties of the modified chitosan oligosaccharide in solution were evaluated by surface tension and dye solubilization, finding that the molecules self-assembled when they are above the critical aggregation concentration. The two methods yielded consistent results, showing that the self-assembly was reduced with higher levels of substitution. The antimicrobial activity of the octanyl-succinylated chitosan oligosaccharide (OSA-COS) derivatives against *Staphylococcus aureus*, *Escherichia coli*, and *Fusarium oxysporum f.sp cucumerinum* was investigated by the Oxford cup method. While the acetylated COS derivatives were not significantly effective against either *E coli* or *S. aureus*, they showed significant antifungal activity toward *F. oxysporum* that was superior to that of COS. The modified product was found to form a stable emulsion when mixed with whey protein isolate. The emulsion formed by the highly substituted derivatives have a certain stability and loading efficiency, which can be used for the encapsulation and delivery of astaxanthin.

## 1. Introduction

Astaxanthin (AST) is a member of the xanthophyll carotenoid family which differs from other carotenoids in having ketone inclusion, together with a terminal hydroxyl in the ionone ring, leading to greater polarity and the presence of antioxidant activity. AST is documented to prevent photo-oxidation by UV light, with ten-fold higher efficiency compared with lutein, β-carotene, canthaxanthine, and zeaxanthin and between 100 and 500 times the antioxidant capability of α-tocopherol [1]. AST has also been shown to enhance neural stem cell proliferation and osteogenic and adipogenic differentiation [2].

Chitosan oligosaccharide (COS) is formed by the hydrolysis of chitin and chitosan. It is more soluble than both parent compounds as a result of its shorter chains and the presence of free amino groups in D-glucosamine moieties [3]. COS is smaller and has reduced viscosity compared with chitosan. The reactive sites in COS consist of two hydroxyl groups in the 3- and 6-carbon positions, respectively, and an amino group at carbon 2 that is susceptible to derivatization. COS is widely used in biomaterials [4] due to its antioxidant, anti-inflammatory, and antifungal properties [5,6,7], leading to its potential use as a food material [8]. Amino derivatives of COS are soluble in water, and aminoethylated COS derivatives were found to inhibit the proliferation of AGS cells [9]. Karagozlu, Karadeniz, and Kong et al. also reported that both COS and its amino derivatives promoted apoptosis through upregulating the levels of Bax and caspases in mitochondrial pathways. Karagozlu et al. [9] also suggested that COS and amino-derivatized COS derivatives induced apoptosis in AGS cancer cells through upregulating the levels of Bax and caspases in mitochondrial pathways. COS can also be modified with Kojic acid [10], fumaric acid [11], and monomethyl fumaric acid (MFA) [12]. These compounds all have good antimicrobial activity, as well as being non-toxic, and can thus be used as food preservatives.

Various food properties, such as stability and texture, are significantly influenced by interactions between proteins and polysaccharides [13]. Thus, protein–polysaccharide complexes can be used for the formation of specific delivery systems by the encapsulation and protection of bioactive substances such as vitamins, probiotics, essential oils, nutraceuticals, and flavors [14,15,16,17,18,19,20]. To date, studies have tended to focus on interactions between proteins from either milk, such as whey and casein proteins, or plants, including proteins from peas and soybeans, non-starch polysaccharides and gelatin, such as pectin, locust bean gum, and κ-Carrageenan [4,21,22,23,24]. There have been very few investigations into interactions between OSA-modified polysaccharides and proteins. Apart from the specific properties of whey protein, the properties of the OSA-modified starch, such as molecular weight (MW) and degree of substitution (DS), also influence complexation; for instance, the DS of OSA-modified polysaccharide has been shown to affect the number of protein-interacting molecules [25,26]. The present study aims to investigate the synthesis and characterization of various octenyl succinylated COS derivatives (OSA-COS) from COS of differing molecular weights. The effects of DS on the interactions between OSA-COS and whey protein in the encapsulation of AST were also evaluated, specifically considering the effects of lipid droplet size on OSA-COS emulsion formation and the bioaccessibility of AST. These findings may assist the development of delivery systems for lipophilic bioactive compounds in foods.

## 2. Materials and Methods

### 2.1. Materials

COS was provided by the Dalian Institute of Chemcial Physics, Chinese Academy of Sciences. Octenyl succinic anhydride (OSA) was purchased from Sigma-Aldrich; methanesulfonic acid and ammonium hydroxide solution were purchased from the Macklin Biochemical Co., Ltd. (Shanghai, China); whey protein isolate powder (WPI), sodium dodecyl sulfate, and Astaxanthin were purchased from Shanghai Yuanye Bio-Technology Co., Ltd. (Shanghai, China); pepsin and lipase were purchased from Shanghai Aladdin Biochemical Technology Co., Ltd. (Shanghai, China); and cholic acid sodium hydrate was purchased from Sinopharm Chemicai Reagent Co., Ltd. (Shanghai, China). Hydrochloric acid, sodium hydroxide, acetone, absolute ethanol, cyclohexane, dimethyl sulfoxide (DMSO-*d6*), and potassium bromide were obtained from Tianjin Kemiou Chemical Reagent Co., Ltd. (Tianjin, China) and Sudan IV from Beijing Solarbio Science & Technology Co., Ltd. (Beijing, China). Professor Qiu Liu of Dalian Minzu University kindly provided the *E. coli*, *S. aureus*, and *Fusarium oxysporum f.sp cucumerinum* (FOC) used in the antimicrobial experiments.

### 2.2. Preparation of OSA-Modified Chitosan Oligosaccharide (OSA-COS)

To avoid the effect of the COS amino group on the subsequent reaction, the COS was reacted with methanesulfonic acid prior to being grafted with OSA. The chitosan derivatives were prepared using 8.1 g of COS with 15 mL methanesulfonic acid for 60 min in an ice-water bath with stirring. COS and OSA were mixed in ratios of 1:6, 1:8, and 1:10. The pH of the sample was held constant at 8.3–8.5 by addition of 1–10% NaOH. Then, the solution was stirred at 25 °C for 6 h and poured into a 3% (*w*/*w*) solution of HCl to form a precipitate which was then dissolved in deionized water. This solution was stirred at 25 °C before addition to aqueous ammonia while maintaining the pH at 6.0 to form a precipitate after centrifugation. The precipitate was washed with cyclohexane by Soxhlet extraction and was refrigerated for 24 h before freezing and lyophilization to obtain a light-yellow powdery material. The samples were termed OSA-C1, OSA-C2, and OSA-C3.

### 2.3. Characterizations of OSA-Modified Chitosan Oligosaccharide

#### 2.3.1. ^1^H Nuclear Magnetic Resonance (NMR) Spectroscopy

As described in our previous method [27], the ^1^H NMR spectra of OSA-COS were obtained on a 500 MHz NMR spectrometer (Agilent-Varian Mercury plus 400) to confirm its structure. Some 5 mg of samples were dissolved in 1 mL of DMSO-*d6* in a 5 mm-walled NMR tube, and the spectra were measured at 500 MHz at 25 °C.

#### 2.3.2. Fourier-Transform Infrared (FTIR) Spectroscopy

After overnight drying of the OSA-COS samples in a 70 °C oven, 1 mg of material was ground with 100 mg of KBr to a uniform powder, as described in our previous publication [28]. A manual press and pellet die were used to obtain a thin pellet. The samples were analyzed on an IR Affinity-1 FTIR spectrometer (Shimadzu, Nagoya, Japan) between 4000 and 400 cm^−1^. Sixty-four scans were performed at a 4.0 cm^−1^ resolution.

#### 2.3.3. X-ray Diffraction Analysis

XRD patterns were collected over a 2θ range between 5° and 60° using an XRD-6000 X-ray diffractometer (Shimadzu) provided with graphite monochromatized high-intensity Cu-Kα radiation (λ = 0.154 nm, 40 kV, 40 mA). The scan rate was 2°/min.

### 2.4. Critical Aggregation Concentration (CAC)

#### 2.4.1. Dye Solubilization

Following the method described in our previous publication [29], a 1% OSA-COS stock solution was made up and diluted to obtain working solutions of varying concentrations. Sudan IV (10 mg) was then added to 10 mL of sample solution and incubated overnight at 40 °C with shaking. The undissolved dye particles were then removed by filtration with a 0.22 μm filter. Absorbances at 510 nm were then measured using a UV-6100 spectrophotometer (Mapada, Shanghai, China), and the CAC was determined from the point where the absorbance increased.

#### 2.4.2. Surface Tension

Static surface tensions of the different OSA-COS solutions were measured with a surface tension meter (HengPing BZY-1) as described in our previous publication [30]. Measurements were taken in triplicate at a constant temperature of 25 °C ± 1 °C. The CAC was calculated from changes in the slope of the plot of equilibrium surface tension against concentration.

### 2.5. Antimicrobial Activity

*E. coli*, *S. aureus*, and FOC were selected as experimental microorganisms, and the Oxford cup method was used to measure antimicrobial activity [31]. The antimicrobial effects of the samples were determined by observing the diameter of the inhibition circle. *E. coli* and *S. aureus* were inoculated in liquid broth medium using an inoculating loop and incubated on a shaker at 37 °C until reaching logarithmic growth. FOC was inoculated into liquid potato glucose medium with an inoculating needle and placed on a shaker at 25 °C until reaching logarithmic growth. Aliquots of microbial cultures (0.5 mL) were inoculated on solid media and placed in a sterile Oxford cup. COS and its derivatives were dissolved in sterile water to a concentration of 20 mg/mL. Potassium sorbate and streptomycin were used as positive controls with 100 μL of sample solutions added to the Oxford cup and incubated at 37 °C and 25 °C, respectively.

### 2.6. Emulsification Properties

#### 2.6.1. Emulsion Preparation

Whey protein isolate powder (WPI) and OSA-COS stock solution were mixed at a ratio of 25:1 (*w*/*w*) and stirred at 25 °C for 30 min. A total of 0.01 g AST was mixed with 10.0 g of medium chain triglycerides (MCT), and the oil phase was obtained after sonication through using an ultrasound cell disruptor (Shanghai Huxi Industrial Co., Ltd. (Shanghai, China)) for 20 min at 400 W. The emulsion containing WPI (1%, *w*/*w*) and OSA-COS (0.04%, *w*/*w*) was made by adding 8.0 g of the mixture solution and 2.0 g of the oil phase in a 20 mL tube. The material was homogenized (IKA T18) at 24,000 rpm for 3 min. The emulsified samples were termed WPI-OC1, WPI-OC2, and WPI-OC3. The control group was prepared by mixing WPI and COS according to the above method.

#### 2.6.2. Emulsion Particle Sizes and Zeta Potentials

Particle sizes were determined immediately after emulsion preparation using a laser particle size analyzer (NanoSizer 3000 E, Malvern Panalytical Corporation, Malvern, UK) at 25 °C. The background values were determined prior to measurement and were subtracted from the total scattering measurements of the samples. Several drops of the sample were placed in distilled water in the dispersion unit. The pump speed was 2000 rpm; obscuration was in the 10–30% range; and the refractive indices of the dispersing medium and particles were 1.33 and 1.45, respectively. The average of duplicate measurement results was used.

Zeta potentials were evaluated using a zeta potential analyzer (HORIBA SZ-100). Before testing, the sample was diluted in deionized water or the corresponding digestion solution to avoid multiple scattering effects and was placed in a cuvette for determination.

#### 2.6.3. Emulsifying Activity Index (EAI) and Emulsifying Stability Index (ESI)

The EAI and ESI values of the samples were measured as described by Sun et al. [32] with several modifications. Briefly, 50 μL of AXT emulsion was homogenized with 10 mL of 0.1% sodium dodecyl sulfate for 2 min at 15,000 rpm (IKA T18 homogenizer). The A_0_ and A_10_ absorbances were determined after 0 and 10 min at 500 nm in the UV-6100 spectrophotometer. EAI and ESI were calculated as follows:(1)$$\mathrm{EAI}\;\left(m^{2}/g\right)=\frac{2\times 2.303\times A_{0}\times \mathrm{dilution}\;\mathrm{factor}}{C\times \Phi \times 10^{4}}$$
(2)$$\mathrm{ESI}\;\left(\min\right)=\frac{A_{0}}{A_{0}-A_{10}}\times 10$$
where C is the initial concentration of protein (g/mL^−1^) and ϕ is the volume fraction of oil phase.

### 2.7. In Vitro Digestion

A simulated gastrointestinal tract (GIT) model was used to assess the behavior of the emulsions in the GIT.

Gastric stage: Following a slightly modified protocol of Li et al. [33], simulated gastric fluid (2 mg/mL NaCl, 7 mg/mL HCl, and 3.2 mg/mL pepsin) was prepared at 37 °C and added to the AXT-loaded emulsions. After adjustment to pH 2.0, the samples were incubated with shaking (100 rpm) for 2 h to simulate gastric conditions.

Small intestinal stage: The method described by Xia et al. [34] was used with some modifications. Three milliliters of saline solution containing CaCl_2_ (150 mM) and NaCl (10 mM), 7 mL PBS containing 375.0 mg of cholic acid sodium hydrate, 5 mL PBS containing 0.12 g lipase, and 60 mL of samples from the gastric stage described above were placed in a reaction vessel to simulate small-intestinal fluid, which was then heated to 37 °C with shaking at 100 rpm for 2 h during which the sample pH was maintained at 7.0 by addition of NaOH.

### 2.8. Astaxanthin Loading Efficiency

The efficiency of AXT loading was measured using a UV–visible spectrophotometer based on the method of Ahmed et al. [35]. Unloaded AXT in the emulsions was extracted three times by the addition of 2 mL of chloroform into 1 mL of emulsion. The collected chloroform solutions were mixed together and centrifuged at 3000 rpm for 5 min at ambient temperature. The lower layer containing solubilized AXT was collected, and the upper layer was mixed with chloroform for collecting the unloaded AXT. Absorbances at 477 were measured. The AXT standard curve was obtained by assaying a series of standard AXT–chloroform solutions. The AXT loading efficiencies were determined as follows [36].
(3)$$\mathrm{Loading}\;\mathrm{efficiency}\;\left(\%,\;\frac{w}{w}\right)=\frac{\mathrm{total}\;\mathrm{astaxanthin}\;-\mathrm{unloaded}\;\mathrm{astaxanthin}\;}{\mathrm{total}\;\mathrm{astaxanthin}}\times 100\%$$

## 3. Results and Discussion

### 3.1. Synthesis and Characterization of OSA-COS

The OSA-COS samples with various DS were successfully synthesized by the addition of OSA to the COS backbone. The detailed procedure is shown in Figure 1a.

The ^1^H NMR spectra of COS and OSA-COS are shown in Figure 1b. In the ^1^H NMR spectrum of COS, peaks ranging from 2.75 to 4.28 ppm indicate the chemical shifts of H on the glucosamine and N-acetylglucosamine residues on the COS sugar ring [10]. The resonance peak of the solvent (DMSO-*d6*) is at 2.49 ppm. The ^1^H NMR spectra of OSA-COS that they have similar profiles. The new characteristic peaks at 0.82, 1.21, and 1.90 ppm signals are assigned to the protons on the methyl and methylene groups of OSA [29]. Furthermore, the peak at 5.36 ppm was attributed to OSA substitution [37]. The presence of the typical peaks of OSA thus confirms the successful synthesis of the OSA-modified COS. The amount of alkyl chains incorporated into the modified samples was calculated from the ratio of the area of the peak at 0.82 ppm to the area of the peaks from 2.75 to 4.28 ppm. The degrees of substitution (DS) determined from the ^1^H NMR spectra are shown in Table 1. The DS increased from 0.14 to 0.19 as the molar ratio of COS to OSA changed from 1:6 to 1:10, indicating that the DS can be increased by augmenting the amount of OSA.

The FTIR spectra of COS, OSA-C1, OSA-C2, and OSA-C3 are shown in Figure 1c. In the COS spectrum, the large band at 3450 cm^−1^ represents -OH and -NH_2_ stretching vibrations, as well as hydrogen bonding, both intra- and inter-molecular, of COS molecules [38]. The bands at 1650, 1560, and 1409 cm^−1^ represent the first, second, and third amide absorption modes [39,40], respectively. Peaks associated with the saccharide backbone include the absorption at 1157 cm^−1^, representing anti-symmetric stretching of the C–O–C moiety, and 1074 cm^−1^ and 1016 cm^−1^, assigned to skeletal vibrations caused by C–O stretching [41]. The OSA-COS spectra show the presence of two new peaks at 1575 cm^−1^ and 1730 cm^−1^, resulting from the formation of ester linkages [28]. The intensity of the band at 2926 cm^−1^ indicating the stretching of –CH_2_– groups has increased, consistent with the previous findings [29]. These results imply that the OSA-COS was successfully synthesized.

The crystalline properties of the materials can be identified by XRD patterns. Figure 1d shows the XRD patterns of pure COS and its OSA derivatives. The COS diffractogram contains two major crystalline peaks located at around 9° and 22° [42]. The OSA crystal structure is characterized by the presence of strong hydrogen bonding (both inter- and intra-molecular) between amino and hydroxyl groups [43]. Reduced intensity of the 2θ = 9.0° peak, together with a reduced maximum value (to 7.5°), can be seen in the OSA-COS spectrum. This reduced intensity may be associated with reduced crystallinity resulting from OSA incorporation into the biopolymer backbone, decreasing the number of free hydroxyl sites for water binding, as crystal Form I is related to a hydrated one [44]. A new peak was visible at 32° in the OSA-COS spectrum. This shift in the position of the peak is the result of increased interlayer distance, resulting from the grafting process.

### 3.2. CAC

The absorbance values obtained for OSA-modified COS after Sudan IV incubation are shown in Figure 2a. It is apparent that the absorbance began to increase when the sample concentration was higher than the CAC, which can be explained by the aggregation of micelles containing the dissolved Sudan IV within their hydrophobic cores [45]. The OSA-COS CAC also declined as the DS increased. As the DS increased, the CAC of OSA-COS became smaller with 6.5 × 10^−5^%, 6.0 × 10^−5^%, and 4.0 × 10^−5^%, respectively. This could be explained by the introduction of OSA increasing the hydrophobicity of OSA-COS, leading to increased self-assembly in aqueous solutions, as previously reported [27,28].

The surface tensions of the OSA-COS sample solutions are shown in relation to concentration in Figure 2b. The OSA-COS samples were able to lower the water surface tension to 49.6 mN/m, 49.8 mN/m, and 57.2 mN/m, respectively, with DS increased from 0.14 to 0.19. In addition, the DS of OSA-COS was positively correlated with surface tension. The inflection point of the curve is the critical aggregation concentration of the sample, and the results are very similar to those obtained by addition of the hydrophobic dye which are 6.5 × 10^−5^%, 6.0 × 10^−5^%, and 4.0 × 10^−5^%, respectively. Table 1 lists the CAC values, indicating their correspondence with the results of the dye solubilization. The data show that highly substituted samples tended to form micellar aggregates at lower concentrations than samples with less substitution, which is consistent with the data provided from the suppliers [28]. A possible explanation is that COS molecules with higher levels of substitution contain more octenyl chains, while the spacing of the octenyl moieties along the COS molecule may also play a part.

### 3.3. Emulsion Properties

The particle sizes and zeta potentials for emulsions prepared with COS with varying DS were assessed shortly after their formation and are shown in Figure 3. As the OSA-COS DS values increased, the volume mean diameter d_(4,3)_ values of the emulsions decreased from 140.9 µm to 89.3 µm. It was found that the OSA-COS samples with varying DS values produced emulsions with smaller particle sizes than unmodified COS. In addition, it was found that the particle sizes of the modified COS emulsions became smaller as the DS values increased. These findings echo those of Kokubun et al. [46], which showed that the droplet sizes of modified inulin emulsions declined as the DS increased.

The zeta-potentials of the four emulsions were −39.2 mV, −39.0 mV, −40.9 mV, and −41.4 mV, respectively, and the potential of the emulsion WPI-OC3 formed by WPI with OSA-C3 was the largest. This increase may have been caused by the increased numbers of negatively charged groups adsorbed on the droplet surface. This increase in surface charge would induce the mutual repulsion between the emulsion droplets, which is beneficial to reduce flocculation and aggregation between droplets and to the stability of the oil–water interface.

Emulsibility includes both EAI and ESI and represents the capacity of water and oil to form an emulsion. The EAI represents the area (m^2^) of the oil–water interface per unit mass of protein (g) stabilizing the interface during mixing. The ESI indicates the capacity of a protein to maintain the emulsion and prevent its separation [32]. Figure 4 shows that both emulsification and its stability were enhanced with increasing DS of the OSA-COS when the protein concentration remained constant and is a factor for determining the emulsion particle size. Due to the particle sizes of the modified COS, emulsions became larger as the DS values decreased from previous results. Larger particle sizes tend to slow the diffusion of the compound to the interface, reducing emulsion formation [47].

### 3.4. Antimicrobial Activity of OSA-COS

In this study, *E. coli, S. aureus*, and FOC were used to investigate the antimicrobial activity of OSA-COS using the Oxford cup method. As shown in Figure 5 and Table 2, the OSA-COS acylated derivatives had no noticeable antibacterial effect on *E. coli* and *S. aureus* but had a significant antifungal effect on FOC, which was better than COS only. Similar to our study, COS-O-terpineol showed minimal antibacterial effect against *E. coli* [48]. The antibacterial effect of viscose fiber–silkworm pupa COS against *S. aureus* was not significant either [49]. The result in this study demonstrates that the introduction of OSA could promote the antimicrobial actions of COS. It further confirms that an important property of the chitooligosaccharide derivatives is their antimicrobial activity against pathogenic bacteria [31].

### 3.5. In Vitro Digestion of the Emulsions

The curves in Figure 6 indicate the percentages of free fatty acid (FFA) release from the emulsion of whey isolate and OSA-COS with varying DS values following digestion of triacylglycerols under small-intestinal conditions by lipase (Figure 6). Pancreatic lipases in the small intestine are responsible for the hydrolysis of triacylglycerol to FFAs and monoacylglycerols. The FFA-release curves were similar in both the experimental and control emulsions, increasing significantly during the first 10 min and then declining, ultimately leveling out at approximately 100 min after which no release occurred. This is consistent with the previous findings [50] and may represent the rapid hydrolysis of phospholipids promoted by bile salts. However, the FFA products tend to inhibit phospholipid breakdown by preventing entry of the pancrelipase or pancreatin to the lipid interface, slowing FFA release over the digestion period [51]. The percentage of FFAs released from the emulsions also increased with increasing OSA-COS DS under the same experimental conditions.

It was observed that the loading efficiencies of AXT in the emulsions prepared with WPI and OSA-COS with different DS were 74.19%, 79.43%, and 86.65%, respectively, which were higher than that of WPI alone (44.28%) (see Figure 7). The loading efficiency was found to be significantly positively correlated with the DS. This may be because of the increased numbers of alkenyl chains on the sugar backbone, resulting in increased intramolecular and intermolecular associations and providing a suitably hydrophobic niche for AXT.

## 4. Conclusions

This study demonstrates the successful modification of COS by OSA in aqueous solutions, resulting in derivatives with varying DS values (0.14, 0.16, and 0.19) and forming micellar-like aggregates. CAC values were observed to decline as the DS increased. As shown by dye solubilization, the OSA-COS aggregates could accommodate hydrophobic compounds within their cores. The acylated COS derivatives showed better antimicrobial activity against FOC than against *E. coli* and *S. aureus*. It was also found that OSA-COS prepared with WPI produced stable emulsions with AXT, with smaller droplet sizes than those formed with WPI only. The amount of FFA released from the emulsions was observed to increase with increasing DS. This, together with the reductions in CAC and particle sizes in response to increasing DS, led to increased solubility and loading efficiency of AXT. These emulsions thus have potential for the encapsulation and delivery of poorly soluble substances in the food, cosmetic, and pharmaceutical industries.

## Figures and Tables

**Figure 1 foods-12-02898-f001:**
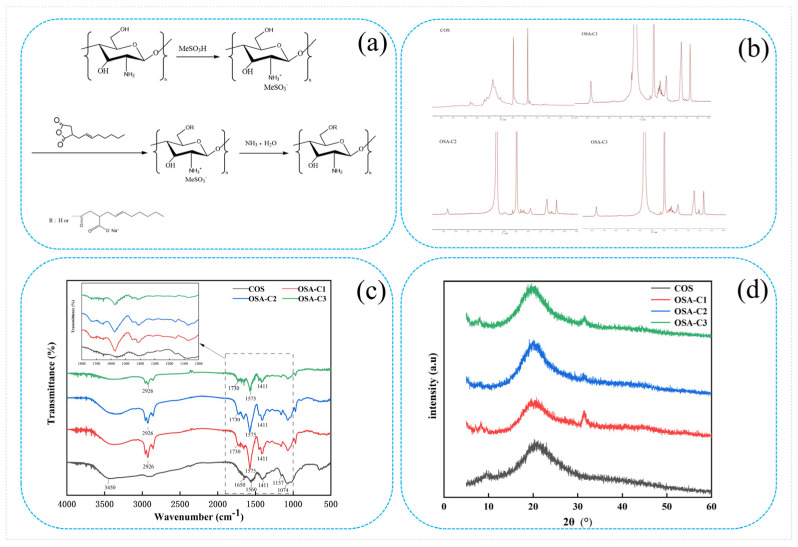
(**a**) Synthetic route of OSA-modified chitosan oligosaccharide; (**b**) ^1^H NMR spectra of COS and OSA−COS; (**c**) FTIR spectra of COS and OSA−COS; (**d**) XRD patterns of COS and OSA−COS.

**Figure 2 foods-12-02898-f002:**
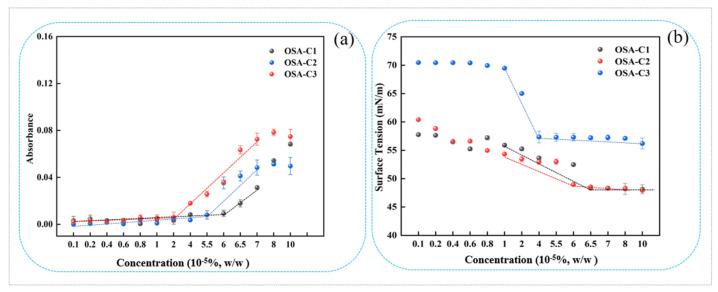
(**a**) Absorbances (510 nm) of OSA-C1, OSA-C2, and OSA-C3 samples after incubation with Sudan IV; (**b**) Surface tension of OSA-C1, OSA-C2, and OSA-C3 in relation to concentration.

**Figure 3 foods-12-02898-f003:**
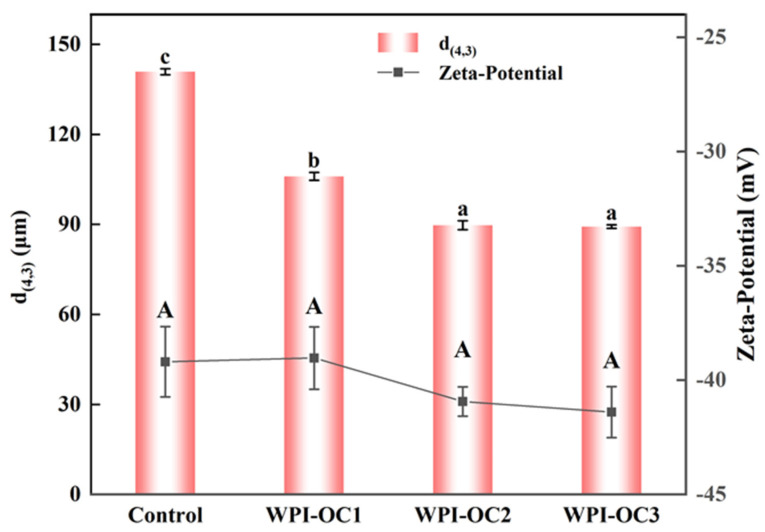
Particle sizes and zeta potentials of astaxanthin emulsions prepared by OSA with different degrees of substitution (Control: unmodified COS emulsion, The emulsified samples were termed WPI-OC1, WPI-OC2, and WPI-OC3). The letter on each column means the significant difference.

**Figure 4 foods-12-02898-f004:**
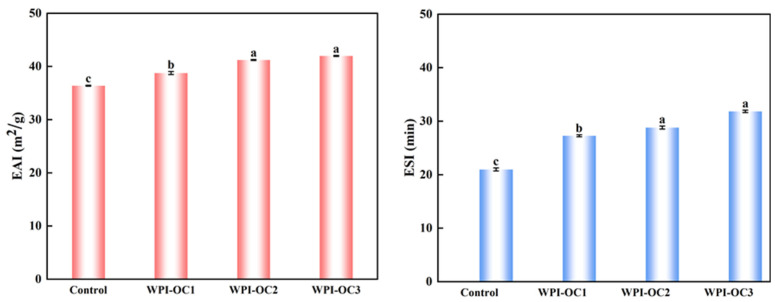
Effects of the degree of substitution on the emulsifying properties of astaxanthin emulsion (Control: unmodified COS emulsion, The emulsified samples were termed WPI-OC1, WPI-OC2, and WPI-OC3). The letter on each column means the significant difference.

**Figure 5 foods-12-02898-f005:**
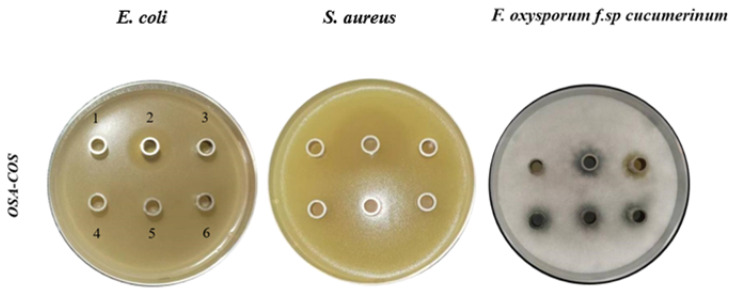
Antimicrobial effects of acylated COS derivatives on *E. coli*, *S. aureus*, and *Fusarium oxysporum f.sp cucumerinum*. (1): sterile water; (2): positive control potassium sorbate for *E. coli* and *Fusarium oxysporum f.sp cucumerinum*, streptomycin for *S. aureus*; (3): COS; (4–6) OSA–COS.

**Figure 6 foods-12-02898-f006:**
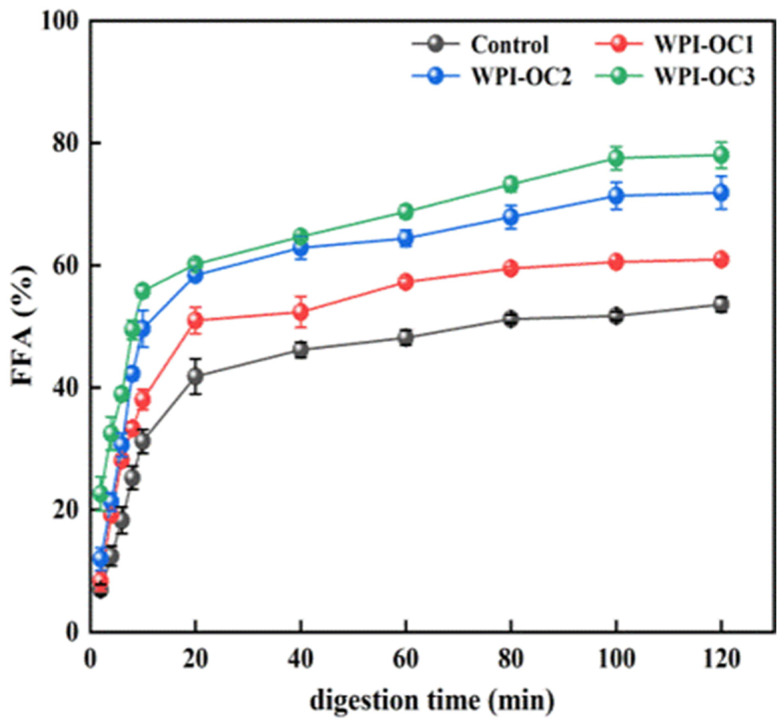
Free fatty acid release from emulsions during simulated small intestinal digestion (Control: unmodified COS emulsion).

**Figure 7 foods-12-02898-f007:**
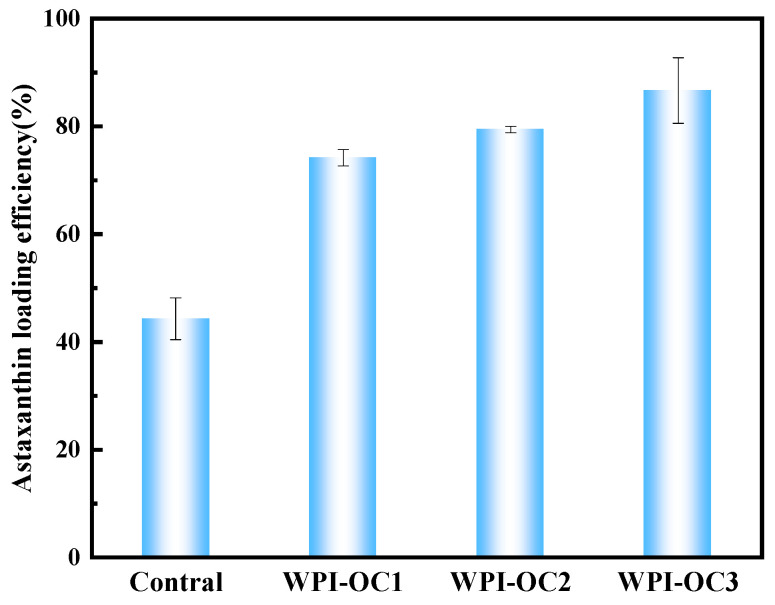
Astaxanthin loading efficiency (Control: unmodified COS emulsion).

**Table 1 foods-12-02898-t001:** Degrees of Substitution of COS.

Sample	Mol (OSA:COS)	DS	CAC (10^−5^%)
OSA-C1	1:6	0.14	6.5
OSA-C2	1:8	0.16	6.0
OSA-C3	1:10	0.19	4

**Table 2 foods-12-02898-t002:** Antibacterial activity of OSA-COS against bacterial test strains.

Sample	Bacteria Name/Inhibition Zone (mm ± SD)		
	*E. coli*	*S. aureus*	*Fusarium oxysporum f.sp cucumerinum*
potassium sorbate	27.1 ± 0.66		17.5 ± 0.57
streptomycin		45.7 ± 0.66	
COS	NA	NA	15.6 ± 0.33
OSA-C1	NA	NA	13.1 ± 0.09
OSA-C2	NA	NA	14.3 ± 0.04
OSA-C3	NA	NA	13.5 ± 0.09

NA: not active. SD: standard deviation of three trials.

## Data Availability

Data are contained within the article.

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
