# Peer review of "Effects of Octenyl-Succinylated Chitosan—Whey Protein Isolated on Emulsion Properties, Astaxanthin Solubility, Stability, and Bioaccessibility"

_foods, 2023, doi:10.3390/foods12152898_

Round 1

Reviewer 1 Report

The paper " Influence of Octenyl-Succinylated Chitosan-Whey Protein-Isolated Emulsion Properties on Astaxanthin Solubility, Stability, and Bioaccessibility" presents the synthesis of octenyl-succinylated chitosan with different degrees of substitution by chemical modification of chitosan. Authors also investigated controlled addition of octenyl succinic acid to chitosan. The authors used appropriate physicochemical methods to study the obtained materials, the results are interpreted correctly. I do not fully understand why such important results as NMR, FTIR and XRD have been placed in the supplementary data. In my opinion, in the case of the FTIR spectrum, it would be clearer to place an additionally enlarged fragment of the range from about 1000 to 1800 cm-1, overlapping the individual spectra one on top of the other. Below are some comments:
•    Line 24: "The emulsion has excellent stability…." I don't think that statement is based on facts.
•    Line 77, 166: I couldn't find an explanation of the abbreviation SDS in the paper
•    Line 144: "oteinWhey pr isolate (WPI) and…." Please correct.
•    Line 145-146: "This milligrams of AST were mixed with 10 g of 145 medium chain triglycerides (MCT) and the oil phase was obtained after sonication." Please standardize the accuracy of the units throughout the work Once we have 10 g once with the accuracy of the first decimal place ..... Or put information with what accuracy the substances were weighed. Next, what apparatus was used for sonication and for how long?
•    I also don't understand the order of the subsections in Results and discussion. Isn't it better to first discuss the whole physicochemistry and then microbiology and in vitro research? There is also no comparative discussion of the results obtained in individual studies.
•    And one last note. The subject matter is very interesting and in recent years a lot of papers have been published on topics related to chitosan, emulsions, functionalization…. However, the literature is a bit poor with recent works. Only 10 of the 51 items are dated 2020 and beyond.

To sum up, I believe that there are numerous minor flaws and factual errors that the Authors should correct.

Author Response

Responses to reviewers’ comments

Comments:

The paper "Influence of Octenyl-Succinylated Chitosan-Whey Protein-Isolated Emulsion Properties on Astaxanthin Solubility, Stability, and Bioaccessibility" presents the synthesis of octenyl-succinylated chitosan with different degrees of substitution by chemical modification of chitosan. Authors also investigated controlled addition of octenyl succinic acid to chitosan. The authors used appropriate physicochemical methods to study the obtained materials, the results are interpreted correctly. I do not fully understand why such important results as NMR, FTIR and XRD have been placed in the supplementary data. In my opinion, in the case of the FTIR spectrum, it would be clearer to place an additionally enlarged fragment of the range from about 1000 to 1800 cm-1, overlapping the individual spectra one on top of the other.

Response: We thank the reviewer for the comments. The 1H NMR, FTIR, and XRD results have been incorporated into the manuscript in accordance with your suggestion. Meanwhile, we place an additionally enlarged fragment of the range from about 1000 to 1800 cm-1, overlapping the individual spectra one on top of the other as your advice. Many thanks for your suggestions which is very helpful. We have addressed the following issues and revised the manuscript accordingly.

Comments:

Q1. Line 24: "The emulsion has excellent stability…." I don't think that statement is based on facts.

Response: We thank the reviewer for pointing this out and have checked throughout the article and made corrections accordingly. Please see the revised abstract Line 24,25,26.

Q2. Line 77, 166: I couldn't find an explanation of the abbreviation SDS in the paper

Response: We thank the reviewer for pointing this out and have checked throughout the article and made corrections accordingly. We use Sodium dodecyl sulfate to instead of SDS. Please see the revised manuscript Line80, 173.

Q3. Line 144: "oteinWhey pr isolate (WPI) and…." Please correct.

Response: We thank the reviewer for pointing this out and have checked throughout the article and made corrections accordingly. Please see the revised manuscript which is marked in red. Please see Line 148.

Q4. Line 145-146: "This milligrams of AST were mixed with 10 g of 145 medium chain triglycerides (MCT) and the oil phase was obtained after sonication." Please standardize the accuracy of the units throughout the work Once we have 10 g once with the accuracy of the first decimal place ..... Or put information with what accuracy the substances were weighed. Next, what apparatus was used for sonication and for how long?

Response: We thank the reviewer for this comment. Base on your suggestion, we have modified the manuscript. “Whey protein isolate powder (WPI) and OSA-COS stock solution were mixed at a ratio of 25:1 (w/w) and stirred at 25°C for 30 minutes. 0.01 g AST were mixed with 10.0 g of medium chain triglycerides (MCT) and the oil phase was obtained after sonication through using an ultrasound cell disruptor (Shanghai Huxi Industrial Co., Ltd.) for 20 min at 400W. The emulsion containing WPI (1%, w/w) and OSA-COS (0.04%, w/w) was made by adding 8.0 g of the mixture solution and 2.0 g of the oil phase in a 20 mL tube. The material was homogenized (IKA T18) at 24 000 rpm for 3 minutes. The emulsified samples were termed WPI-OC1, WPI-OC2, and WPI-OC3. The control group was prepared by mixing WPI and COS according to the above method.”  We have amended the text in the revised manuscript with red. Please see Line 148-156.

Q5. I also don't understand the order of the subsections in Results and discussion. Isn't it better to first discuss the whole physicochemistry and then microbiology and in vitro research? There is also no comparative discussion of the results obtained in individual studies.

Response: We thank the reviewer for this comment. We have jiggled the order of the subsections in Results and discussion. Now the results and discussion of manuscript is the discussion the whole physicochemistry firstly and then microbiology and in vitro research. We have also added comparative discussion of the results, such as “The inflection point of the curve is the critical aggregation concentration of the sample and the results are very similar to those obtained by addition of the hydrophobic dye which are 6.5×10-5%, 6.0×10-5% and 4.0×10-5%, respectively (Please see Line 274-275). Fig. 4 shows that both emulsification and its stability were enhanced with increasing DS of the OSA-COS when the protein concentration remained constant and is a factor to determine of the emulsion particle size. Due to the particle sizes of the modified COS emulsions became larger as the DS values decreased from previous results. (Line 315-319)” Please see the manuscript which is in red.

Q6. And one last note. The subject matter is very interesting and in recent years a lot of papers have been published on topics related to chitosan, emulsions, functionalization…. However, the literature is a bit poor with recent works. Only 10 of the 51 items are dated 2020 and beyond.

Response: We thank the reviewer for this comment. We have use four newly relevant literatures to instead of some literature. Please see Line 420-432.

  1. Zhou, J.; Wen, B.; Xie, H.; Zhang, C.; Bai, Y.; Cao, H.; Che, Q.; Guo, J.; Su, Z., Advances in the preparation and assessment of the biological activities of chitosan oligosaccharides with different structural characteristics. Food Funct 2021, 12 (3), 926-951.
  2. Affes, S.; Aranaz, I.; Acosta, N.; Heras, Á.; Nasri, M.; Maalej, H., Physicochemical and biological properties of chitosan derivatives with varying molecular weight produced by chemical depolymerization. Biomass Conversion and Biorefinery 2022.
  3. Hu, K.; Yuan, X.; He, H.; Zhang, H.; Wang, F.; Qiao, J., Pharmacological mechanisms of chitotriose as a redox regulator in the treatment of rat inflammatory bowel disease. Biomed Pharmacother 2022, 150, 112988.
  4. Li, R.; Zhu, L.; Liu, D.; Wang, W.; Zhang, C.; Jiao, S.; Wei, J.; Ren, L.; Zhang, Y.; Gou, X.; Yuan, X.; Du, Y.; Wang, Z. A., High molecular weight chitosan oligosaccharide exhibited antifungal activity by misleading cell wall organization via targeting PHR transglucosidases. Carbohydr Polym 2022, 285, 119253.

Reviewer 2 Report

Manuscript foods-2495714-peer-review-v1 presents an interesting and well conducted study. Few comments.

The title is a bit confusing. It could be changed.

Line 46. Instead of "These authors", name the authors.

Line 77: Please define SDS.

Line 144: Please correct typo.

Please check and provide all measuring units.

Measurement units to be added in the figure’s axes also (Fig. 1, 2) and Tables (.

3.3. Antimicrobial activity of OSA-COS. Please indicate the diameter of the inhibition circles for all samples (in a Table).

Figures 1 and 2 – Lines (not only dots) to be added.

Author Response

Comments:

Response: We thank the reviewer for the comments. We have addressed the following issues and revised the manuscript accordingly.

Q1. The title is a bit confusing. It could be changed.

Response: We thank the reviewer for this comment. We have changed the title. The new title is Effects of Octenyl-Succinylated Chitosan-Whey Protein-Isolated on Emulsion Properties, Astaxanthin Solubility, Stability, and Bioaccessibility.

Q2. Line 46. Instead of "These authors", name the authors.

Response: We thank the reviewer for this comment. We have shown the name of authors which are Karagozlu, Karadeniz, and Kong, et al. Please see Line 47,48.

Q3. Line 77: Please define SDS.

Response: We thank the reviewer for pointing this out and have checked throughout the article and made corrections accordingly. We use Sodium dodecyl sulfate to instead of SDS. Please see the revised manuscript Line80, 173.

Q4. Line 144: Please correct typo.

Response: We thank the reviewer for pointing this out and have checked throughout the article and made corrections accordingly. Please see the revised manuscript which is marked in red. Please see Line 148.

Q5. Please check and provide all measuring units.

Response: We thank the reviewer for this comment. We have checked and provided all modified units in the manuscript.

Q6. Measurement units to be added in the figure’s axes also (Fig. 1, 2) and Tables.

Response: We thank the reviewer for this comment. We have added the measurement units in the figure’s axes and tables. Because of the modification of the manuscript, please see Figure 2 which is the original Figure 1 and Figure 2.

Q7. 3.3. Antimicrobial activity of OSA-COS. Please indicate the diameter of the inhibition circles for all samples (in a Table).

Response: We thank the reviewer for this comment. We have indicated the diameter of the inhibition circles for all samples, please see Table 2.

Q8. Figures 1 and 2 – Lines (not only dots) to be added.

Response: We thank the reviewer for pointing this out and we have added lines on the figures. Please see Figure 2 due to the modification of the manuscript.